# The Depth of Water Taken up by Walnut Trees during Different Phenological Stages in an Irrigated Arid Hilly Area in the Taihang Mountains

**Yang Liu [1], Xuemei Zhang [1,2,3], Shuang Zhao [4], Huabing Ma [2,3], Guohui Qi [1,2,3,*] and Suping Guo [1,2,3]**

[1]  College of Forestry, Hebei Agricultural University, Baoding 071000, Hebei, China; wolongchina@163.com (Y.L.); zhangxuemei888@163.com (X.Z.); lbg888@163.com (S.G.)
[2]  Research Center for Walnut Engineering and Technology of Hebei, Lincheng 054300, Hebei, China; 15128488314@163.com
[3]  Institute of Walnut Industry Technology of Hebei Province (Xingtai), Lincheng 054300, Hebei, China
[4]  Hebei Academy of Agriculture and Forestry Sciences, Shijiazhuang 050000, Hebei, China; zhaoshuang0823@sina.com
*  Correspondence: bdqgh@sina.com; Tel.: +86-1351-328-1897

**Abstract:** Understanding how the soil environment impacts root water uptake location and magnitude is important for better management of plant irrigation. In this study, stable hydrogen and oxygen isotope composition were used to determine seasonal variations in the depth of water taken up by walnut trees during different phenological stages in an irrigated arid hilly area in the Taihang Mountains in China. The contributions of soil water at different depths to the water taken up were quantified by the MixSIAR Bayesian isotope mixing model. The results indicated that water taken up by the walnut trees was sourced mainly from soil water in the 0–20 cm soil layer at the sprouting and leaf expansion stages (62.95%), and the 20–40 cm soil layer at blossoming and fruit-bearing (43.45%), fruit expansion (41.8%), and fruit maturity (39.15%) stages. The mean soil depth of the water taken up by the walnut trees gradually decreased as the phenological stages advanced. The proportions of various soil layer water contributions to the walnut trees differed throughout the phenological stages, and the proportion of deeper soil water contributions gradually increased as the phenological stages of walnut trees advanced. The results of the present study indicated that water sources for walnut trees varied by depth during different phenological stages. In addition to soil moisture, soil temperature may also be an important factor affecting the depth of water taken up by walnut trees. The results also provided scientific implications for water-saving irrigation management.

**Keywords:** water sources; stable isotopes; MixSIAR model; walnut; phenological stage

## 1. Introduction

Walnut (*Juglans regia* L.) is a deciduous tree in the walnut family and is the source of one of the famous "four nuts" throughout the world. Walnut forests provide several ecological and economic benefits [1]. Walnut cultivation has become an important pillar industry for the improvement of the economic situation of mountain farmers, and walnut has been listed as a national strategic economic forest tree by the State Forestry Administration in China. The yield and quality of walnut depend directly on the water supply status [2]. The root system is the main carrier of material cycles and energy flow in both vegetation and soil. Root depth and distribution determine the potential source of soil moisture for plant absorption [3,4]. In the air-soil-vegetation system, the function of plant roots in the water cycle is difficult to assess. It is always difficult to study the source of water in plants by

traditional methods [5]. Stable hydrogen and oxygen isotopes are considered the "fingerprints" of water. The use of these isotopes provides a new means of observation for studying water use and distribution in plants [6].

Water absorbed by plant roots is not usually subjected to hydrogen–oxygen isotope fractionation before being transported to young tender branches or leaves through the xylem of stems [7,8]. Therefore, the hydrogen and oxygen isotopes in the water in plant stems and potential water sources can be compared via linear mixing models [9] or the IsoSource mixing models [10,11], and the ratio of each potential water source that plants use can be calculated [12]. Ma and Song [13] reported that, during the wet season, the depth of water absorption of summer maize was mainly concentrated in the upper soil depth (0–50 cm), and the roots absorbed deeper water during the dry season. The change in soil moisture distribution was in accordance with the variation in the seasonal water absorption pattern. Using stable isotope technology, Liu et al. [14] studied the depth of water taken up by subalpine shrubs in the Wolong Nature Reserve, and the results indicated that these subalpine nonphreatophytic shrubs take up soil water primarily from the top 30 cm in the soil profile. Water uptake patterns were significantly positively correlated with rootlet biomass distribution as well as the soil water content profile. The water absorption depth of plant roots was also closely linked with the range of active root zones [15]. Rothfuss et al. [16] pointed the $\delta^{18}O$ in organic matter at the leaf base was strongly correlated with the $\delta^{18}O$ in stem water, indicating that it could be a good proxy for source water in extensive samplings. At present, research on the use of stable isotopes to study plant water sources focuses mostly on vegetation grown under natural conditions; few studies have investigated the depth of water taken up by trees under irrigated conditions. If we can understand the main soil layer of water taken up by plants in different seasons, according to the theory of partial root-zone drying (PRD) [17], we only need to supply water to the soil layer by using some special equipment, and the plants can grow normally. In this way we can reduce the evaporation of surface soil water or water deep percolation, which could improve water use efficiency.

Therefore, we chose walnut trees at the fruiting stage as the research object and analyzed the depth of water taken up by walnut trees at different phenological stages. In addition, the relationships between the water use strategy of walnut trees and environmental factors were discussed to provide a theoretical basis for the establishment of a scientific-based water-saving irrigation scheme for walnut trees at the fruiting stage.

## 2. Materials and Methods

### 2.1. Site Description

The research was conducted at the 'Lijiahan' walnut demonstration base of Hebei Lvling Fruit Industry Co., Ltd. (114°30′–114°33′ E, 37°29′–37°32′ N, 90–135 m above sea level), which was 6 km north of Lincheng County, Hebei Province, China (Figure 1). The area has a warm temperate and monsoonal climate with four clearly distinct seasons. The mean annual temperature is 13.0 °C, and the minimum and maximum air temperatures are −23.1 °C and 41.8 °C, respectively. The mean annual rainfall is 521 mm. The frost-free period is 202 days. The soil type is cinnamon, and the depth of the soil is approximately 80 cm. Walnut trees (variety 'Lvling') were planted in the spring of 2005; the row spacing was 3 m × 5 m, and the lines ran north–south. During the experiment, the soil moisture status was constantly monitored to determine whether to irrigate. Irrigation was applied on 16 March, 12 May, 21 June and 15 August, and the irrigation amount was 60 kg/tree.

### 2.2. Meteorological Measurements

Weather data, including the daily precipitation, maximum and minimum air temperatures, solar radiation, average wind speed, and average relative humidity, were measured using an automatic weather station (model TYD-ZS1, Tianyude, Beijing, China). The soil volumetric water content and

temperature were measured in 20 cm increments (0–20, 20–40, and 40–60 cm) by a soil temperature and moisture recorder (model L99-TWS-3, Loggertech, Hangzhou, China).

### 2.3. Water Sampling and Isotope Analyses

During each phenological stage (the sprouting and leaf expansion stage—from late March to mid-April; the blossoming and fruit-bearing stage—from late April to mid-May; the fruit expansion stage—from mid-June to the end of July; and the fruit maturity stage—from early August to early September), tree and soil samples were collected for water stable isotope analysis. The water samples were collected at the sprouting and leaf expansion stage on 26 March and 15 April; at the blossoming and fruit-bearing stage on 25 April and 15 May; at the fruit expansion stage on 15 June, 1 July, and 24 July; and at the fruit maturity stage on 7 August, 18 August, and 1 September. The data for 2–3 days at each stage were then averaged to represent the values of the stage.

Lignified twigs (no leaves or green tissue were included in the live twig parts) approximately 5–10 mm in diameter and 5 cm long were cut from the base of the canopy in the four cardinal directions (east, south, west, and north) in each sampling tree, and all leaves and green stem tissue were removed from these twigs to avoid contamination of xylem water by isotopically enriched water [7]. Clipped twigs were immediately placed in a capped glass vial, wrapped in parafilm, and placed in a cooler with ice for transportation to the laboratory.

Soil samples beneath each of the sampling trees were collected at depths of 10, 20, 30, 50, and 70 cm, using a soil auger, and were also placed in a capped vial, wrapped in parafilm, and frozen for subsequent analysis.

Irrigation water was collected at each irrigation. The samples were also each placed in a capped vial, wrapped in parafilm, and frozen for subsequent analysis.

A funnel and a polyethylene bottle were connected together as a rain collector. A ping-pong ball was put in the funnel to prevent evaporation [18]. Precipitation samples were collected in airtight glass vials after each rain event.

All samples were frozen in a freezer ($-15\ ^{\circ}$C to $-20\ ^{\circ}$C) before isotopic analysis. Water in the soil samples and walnut stem samples was extracted using an automatic vacuum condensing extraction system (model LI-2100, Lica United Technology Limited Inc., Beijing, China). Isotopic composition of the liquid samples was analyzed with an isotopic ratio infrared spectroscopy system (model L2120-i, Picarro Inc., Santa Clara, CA, USA) [19]. Catalytic oxidation allows in line sample treatment to remove alcohols from plant water samples. All the water samples were calibrated and normalized to internal laboratory water standards that were previously calibrated to the Vienna Standard Mean Ocean Water (VSMOW, 0‰). The results were expressed as δ values, which were relative to the VSMOW on a normalized scale: $\delta D$ or $\delta^{18}O$ (‰) = $(R_{sample} - R_{standard})/R_{standard} \times 1000$, where $R_{sample}$ and $R_{standard}$ are the molar ratios of D/H or $^{18}O/^{16}O$ of the sample and standard water (VSMOW). The analytical errors for D and $^{18}O$ were $\pm 0.4$‰ and $\pm 0.1$‰, respectively.

### 2.4. Data Analysis

The MixSIAR Bayesian isotope mixing model (v3.1) [20] was used to identify sources of water used by walnut trees. In this study, the potential source of water uptake by walnut trees was considered to be soil water at different depths, which was mixed proportionally with old soil water, rainfall, and irrigation. Groundwater was not considered one of the water resources for walnut trees because of the deep water table depth (on average 30 m below the soil surface) in the study area. The input data of the MixSIAR model were the measured dual isotope values ($\delta D$ and $\delta^{18}O$) of stem water and soil water at different layers, and mean isotope discrimination values for each soil water source with standard error. The discrimination values were set to zero for both $\delta D$ and $\delta^{18}O$, mainly because there is no isotope fractionation during plant water uptake [7]. Individual effects as a random occurrence were included in all analyses. The Markov Chain Monte Carlo parameter was set to "normal" run length. The error options of "residual error" and "process error" were specified in the model. Trace plots and

the diagnostic tests Gelman-Rubin, Heidelberger-Welch, and Geweke were used to determine whether the model converged or not. The estimated median (50% quantiles) proportion (the median source contribution value for each water source) was analyzed for comparisons.

The differences among the soil temperature and soil water content (SWC), and contribution of various soil water sources were assessed by a one-way ANOVA using SPSS 21.0 software (SPSS Inc., Chicago, IL, USA). Proportions of soil water contributions to walnut trees were made by R 3.5.1 package (R Development, Core Team, 2018). The pictures were made by OriginPro 2018 (OriginLab Inc., Northampton, MA, USA).

## 3. Results

### 3.1. Rainfall and Temperature

The daily rainfall during the phenological stages of walnut is shown in Figure 1. The total rainfall was 259.0 mm, of which 46.8, 2.0, 151.6, and 58.6 mm fell during the sprouting and leaf expansion stage (from late March to mid-April), blossoming and fruit-bearing stage (from late April to mid-May), fruit expansion stage (from mid-June to the end of July), and fruit maturity stage (from early August to early September), respectively.

A range of temperatures occurred in the walnut orchards during the phenological stages of walnut from early March to the end of September (Figure 1). The temperature increased sharply from early March to early May; however, the increasing temperature trend slowed from early May to mid-July, after which the air temperature started to decrease. The mean temperature was 13.8, 22.5, 27.5, and 25.5 °C at the sprouting and leaf expansion stage, blossoming and fruit-bearing stage, fruit expansion stage, and fruit maturity stage, respectively.

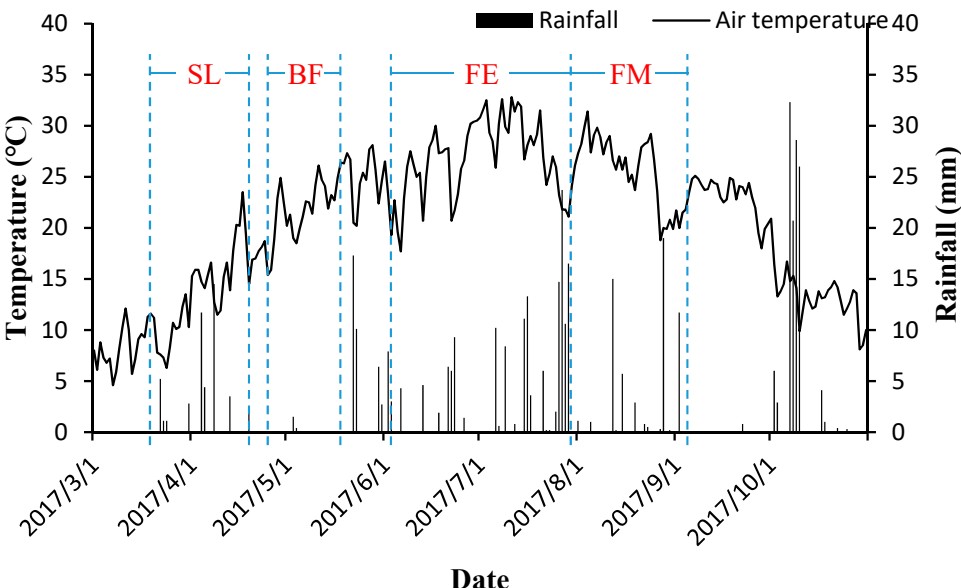

**Figure 1.** Rainfall distribution and temperature from March to October in 2017 (SL: sprouting and leaf expansion stage; BF: blossoming and fruit-bearing stage; FE: fruit expansion stage; FM: fruit maturity stage. The labels are the same below.).



### 3.2. Soil Temperature Conditions

The soil temperature exhibited significant seasonal changes during different phenological stages of walnut, and there were also great differences among different soil layers at the same phenological stage, which were more obvious in the flowering and fruit setting stage and the fruit swelling stage (Figure 2). The mean soil temperature at depths of 0–20, 20–40, and 40–60 cm was 12.5, 11.9, and 11.4 °C at the sprouting and leaf expansion stage, respectively. The mean soil temperature at depths of 0–20, 20–40, and 40–60 cm was 18.7, 17.7, and 16.8 °C at the blossoming and fruit-bearing stage, respectively, and compared with that at the sprouting and leaf expansion stage, the soil temperature at this stage increased sharply at various soil depths. The mean soil temperature at depths of 0–20, 20–40, and 40–60 cm was 24.3, 23.4, and 22.5 °C at the fruit expansion stage, respectively, and compared with that at the blossoming and fruit-bearing stage, the soil temperature at this stage clearly increased. The mean soil temperature at depths of 0–20, 20–40, and 40–60 cm was 23.8, 23.5, and 22.9 °C at the fruit maturity stage, respectively, and compared with that at the fruit expansion stage, the soil temperature at this stage decreased mildly.

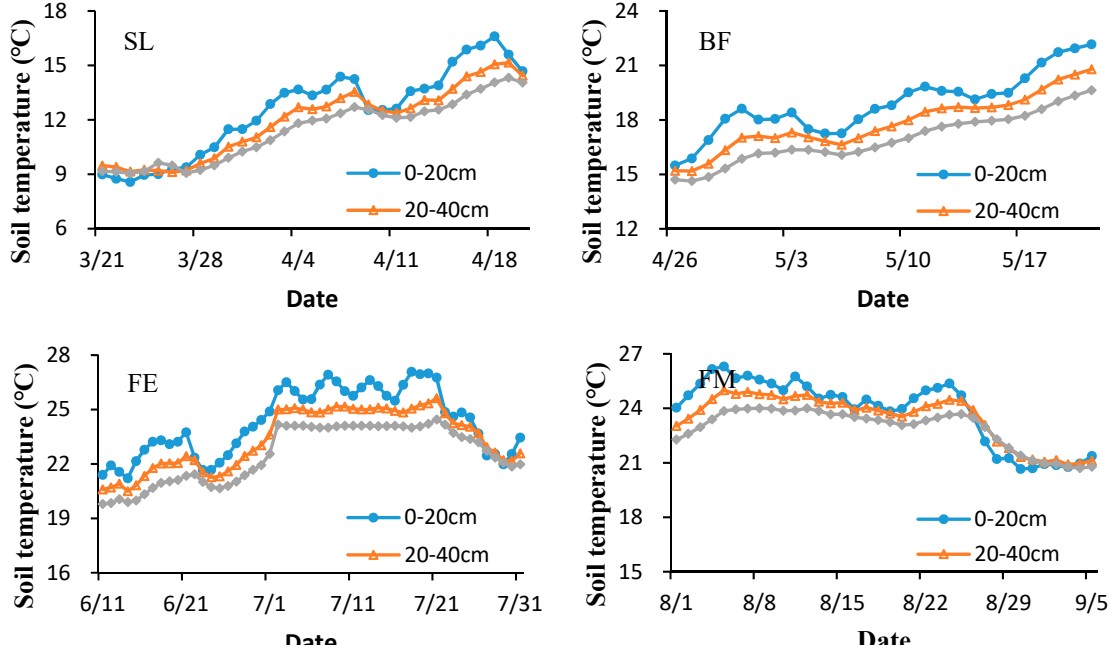

**Figure 2.** Soil temperatures during different phenological stages.

### 3.3. Soil Water Content

The soil water content exhibited significant seasonal changes at different soil depths during the phenological stages of walnut (Figure 3). The SWC in the 0–20 cm soil layer was significantly lower than that in the 20–40 and 40–60 cm soil layers, and the change in temperature in the 0–20 cm soil layer was greater than that in the 20–40 and 40–60 cm soil layers because, compared with the other soil layers, the surface soil was more susceptible to rainfall and evaporation. Due to the application of irrigation water, no significant differences in the average soil moisture were observed in the 0–60 cm soil layer at the other three phenological stages with the exception of the fruit expansion stage. The fluctuations in soil moisture at the fruit expansion stage and fruit maturity stage were clearly greater than those at the sprouting and leaf expansion stage and blossoming and fruit-bearing stage, which were mainly due to the increase in temperature and the uneven distribution of increasing amounts of rainfall.

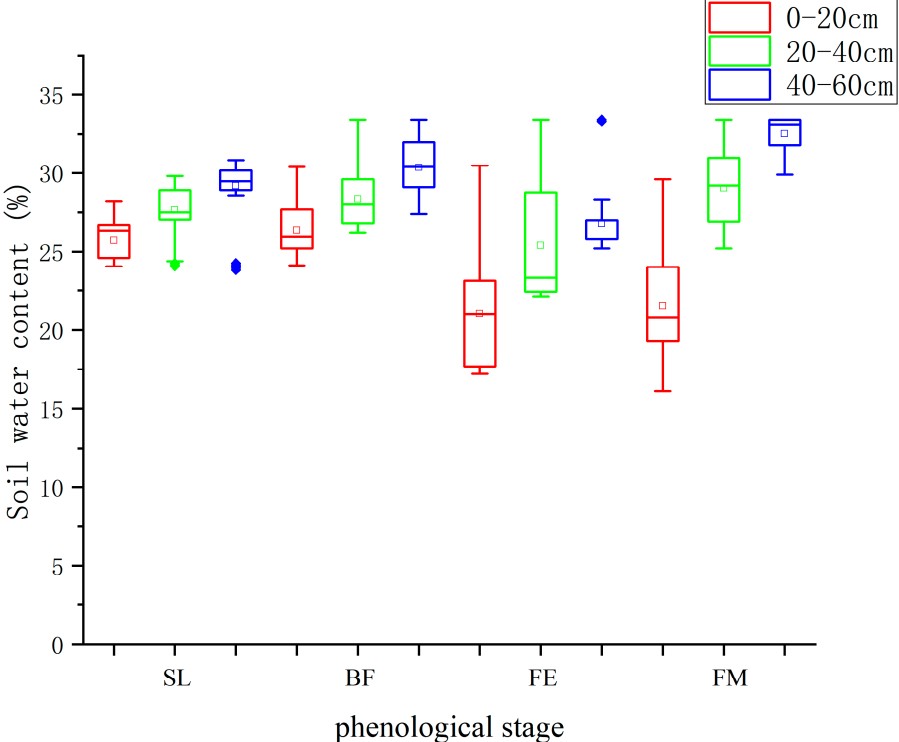

**Figure 3.** The soil water content during different phenological stages.

### 3.4. Isotopic Composition of Water Samples

There was a significantly linear relationship between $\delta^{18}O$ and $\delta D$ in rain water samples ($\delta D = 5.86$ $\delta^{18}O - 12.52$, $R^2 = 0.98$) (Figure 4). The slopes of the local meteoric water line (LMWL) were closed to that of the global meteoric water line (GMWL), expressed as $\delta D = 8$ $\delta^{18}O + 10$ [21]. There was no significant variation in irrigation water samples during different phenological stages, which was ascribed to the fact that irrigation water comes from below 30 m underground. Although rainwater and irrigation water were the initial sources of water in walnut trees (not including groundwater because of its level below 30 m), only after the infiltration process and changing into soil water, could they be absorbed by walnut roots. The measured water in xylem was a mixture of water in different soil layers that could be absorbed by walnut roots, which is why stem water was located in the middle of soil water.

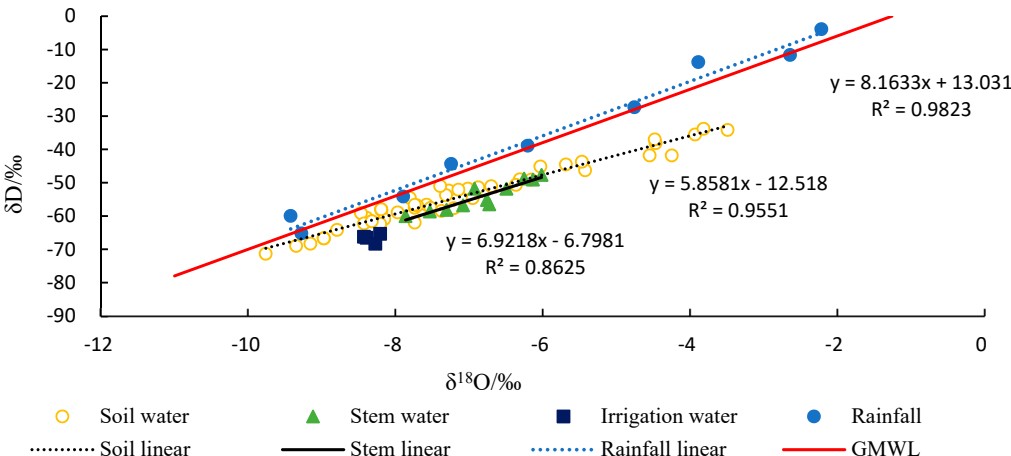

**Figure 4.** Distribution characteristics of hydrogen and oxygen stable isotope ratio ($\delta D$ and $\delta^{18}O$) values of stem water and soil water.

*3.5. Isotopic in Soil Profile and Walnut Stem*

　　　The variation of δD was similar to that of δ$^{18}$O with soil profile depth, but there were also some differences between them (Figure 5). The depths of water uptake can be estimated by the isotopic intersections between xylem water vertical lines and soil water stratum [22]. If there is more than one intersection, the conjunct one (the intersection between 0 and 10 cm) was usually chosen as the main depth of the water source [23]. According to this graphical inference method, the walnut trees derived most of their water from the 0–20 cm soil layer at the sprouting and leaf expansion stage, 10–30 cm soil layer at the blossoming and fruit–bearing stage, 20–40 cm soil layer at the fruit expansion stage, and 20–40 cm soil layer at the fruit maturity stage. In general, the mean soil depth of water taken up by the walnut trees gradually decreased as the phenological stages of walnut advanced.

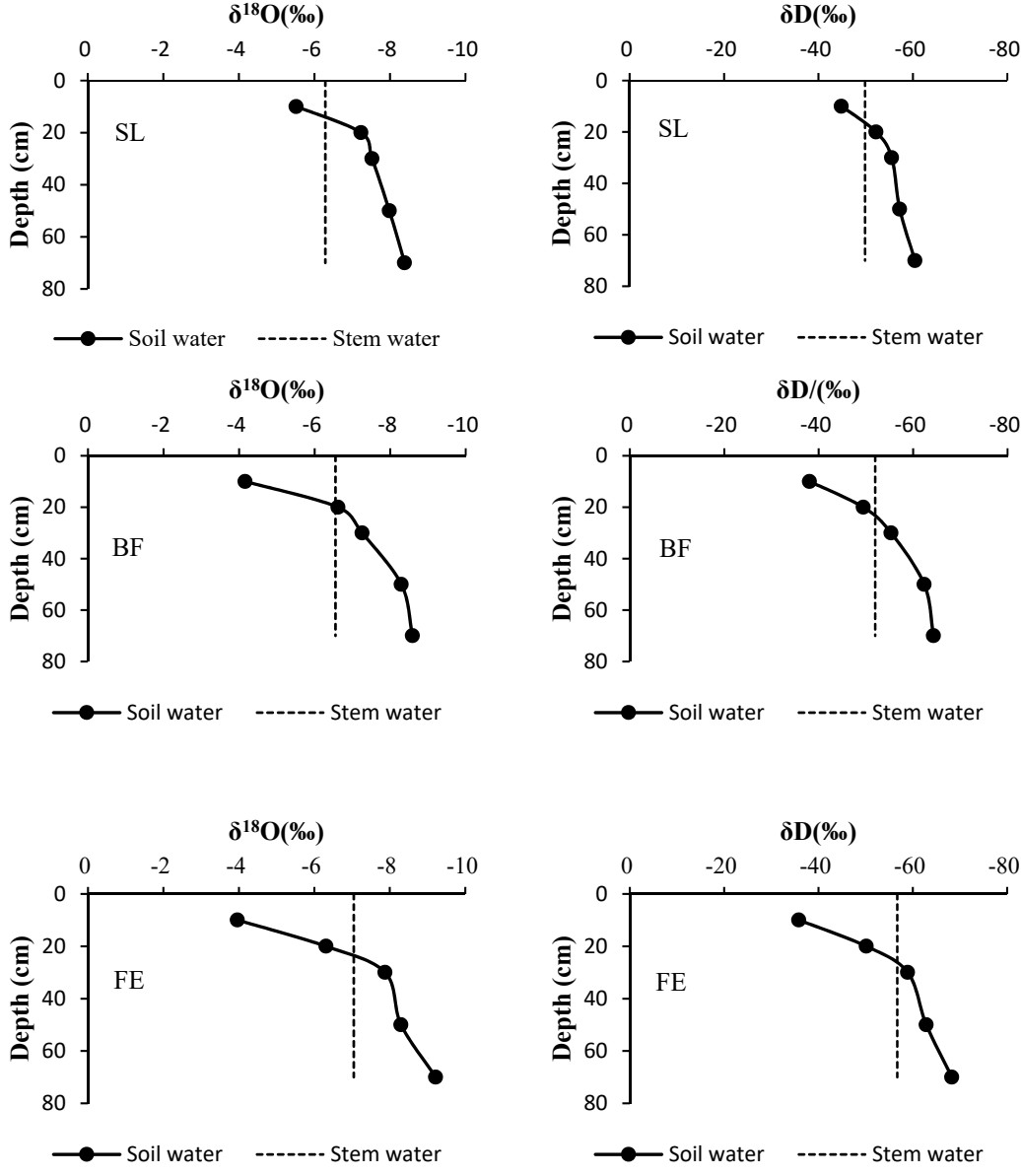

**Figure 5.** *Cont.*

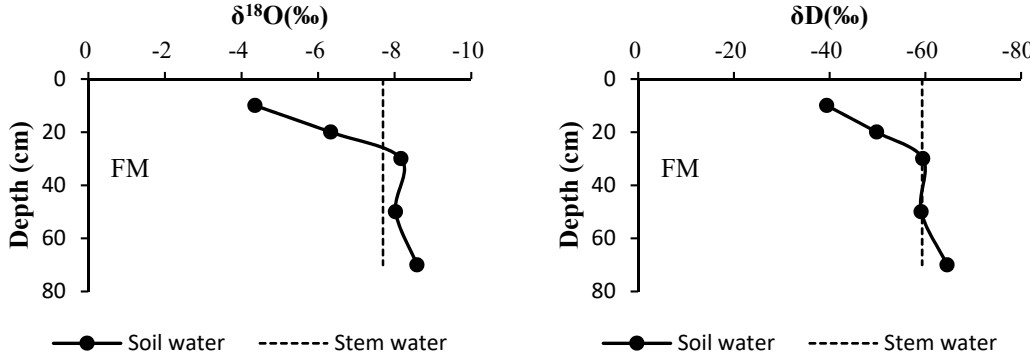

**Figure 5.** The δD and δ¹⁸O values of the soil water and stem water during different phenological stages.

*3.6. Proportions of Soil Water Contributions to Walnut Trees*

The water sources were defined as the soil water in each layer, and the mean isotopic values for each layer represented the potential water sources. Results from the MixSIAR Bayesian isotope mixing model showed that there was a significant difference in the proportions of soil water contributions to walnut trees at different phenological stages, and the proportions of deeper soil water contributions gradually increased as the phenological stages advanced (Figure 6). At the sprouting and leaf expansion stage, the water taken up by the walnut trees was sourced mainly from the 0–20 cm soil layer, the contribution of which reached 62.95%. At the blossoming and fruit-bearing stage, the water taken up by the walnut trees was sourced mainly from the 0–40 cm soil layer, among which the 20–40 cm soil layer accounted for the largest proportion, 43.45%. At the fruit expansion stage, the water taken up by the walnut trees was sourced mainly from the 20–40 cm soil layer (41.8%). Last, at the fruit maturity stage, the water taken up by the walnut trees was sourced mainly from the 20–40 cm soil layer (39.15%), and 0–20 cm soil water contributed much less (12.05%).

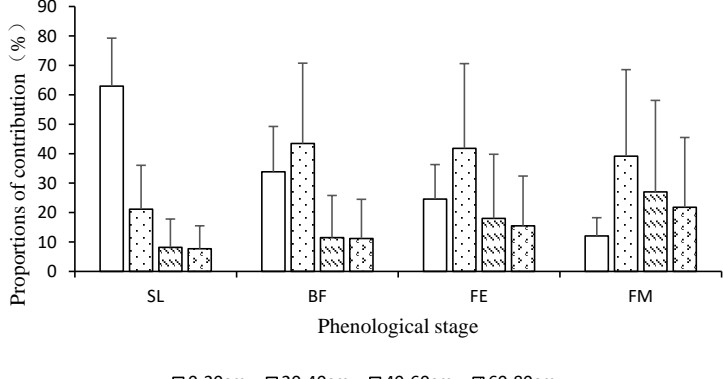

**Figure 6.** Proportions of soil water contributions to walnut trees during different phonological stages.

## 4. Discussion

*4.1. Variations in the Isotopic Composition of Soil Water and Plant Water*

In the study, the isotopic composition of the soil water exhibited significant differences at various soil depths at different phenological stages. The isotopic composition of soil water was affected by environmental factors such as irrigation, precipitation, groundwater, and evaporation [24,25]. Rainfall varied greatly in different seasons and the evaporation was also different because of temperature differences. During the spring, the influence of environmental factors on soil water was low because of low temperature and little rainfall. During the summer, the effect of these factors increased sharply as the temperature and precipitation increased. Compared with deep soil layers, the change of the

SWC and isotopic composition at shallow soil layers throughout the phenological stage was more obvious. The variations in the isotopic composition of the soil water clearly diminished as the soil layer depth increased, which suggests that these variations were greatly influenced by both evaporation and rainfall recharge [26]. A strong evaporation effect generally occurs in shallow soil layers and decreases as the depth increases [27], and the effects of evaporation and soil hydrological processes (e.g., infiltration) gradually weaken as the soil layer depth increases. Our findings are in good agreement with the results of the study by Tang and Feng [28], in which the highly variable isotopic composition of surface soil water was controlled by evaporation enrichment and isotopic mixing between water of different precipitation events. Additionally, Dai et al. [29] reported that soil physical properties differed among various soil layers, which may have affected the extent to which the depth of the SWC and isotopic composition were influenced by precipitation infiltration and evaporation.

There were typical seasonal variations of the isotopic composition in the walnut stems, which were similar to the results of previous studies [13]. Although rainwater and irrigation water were the initial sources of water in walnut trees (not including groundwater because of its level below 30 m), only after the infiltration process and changing into soil water could they be absorbed by walnut roots. The results suggest that those variations depended on the isotopic composition of the soil water and the main soil layer of water absorption by walnut trees. As the phenological stages of walnut advanced, they tended to use water within deeper soil layers; the deeper the soil layer was, the lower the δD and δ$^{18}$O of the soil water. As opposite to our results, Zhou et al. [30] reported that the water within several plants (such as *Tamarix ramosissima* Ledeb.) was sourced from both deep soil water and groundwater, and that no seasonal variations in isotopic composition were observed. These results contrast with those of our experiments, which resulted from differences in both root distribution patterns and habitat environments.

*4.2. Seasonal Water Use Patterns of Trees*

In the present study, the proportions of various soil layer water contributions to plants differed throughout the phenological stages (Figure 6). This phenomenon may explain the typical seasonal variations observed in the isotopic composition of plant xylem water. Zhang et al. [31] also found that there were significant differences in water absorption depth of Maize at different growth stages. The proportion of soil water taken up by the plants generally increased as the SWC increased [32,33]. As the phenological stages of walnut advanced, the SWC at shallow soil layers gradually decreased and shallow roots absorbed less water. However, Wu et al. [15] found that despite the abrupt increases in SWC in shallow soil on 18 September 2013, the water-source depths of *Hippophae rhamnoides* L. did not change, whereas native plants still held similar water-use patterns as in August. The depth of water taken up by plants is ultimately determined by the water absorption of roots in various soil layers. The water absorption by roots depends on the number and water uptake ability of roots. The water absorption rate of the root system in various soil layers is due to the differences in soil environment. The previous result could be better explained as a combination of SWC and soil temperature. Some studies have suggested that the source of water taken up by plants should be determined by considering other environmental parameters, such as root distribution [14], SWC [23], leaf water potential [34], and throughfall precipitation [35]. Zhou et al. [30] found that more than 90% of the water of *Tamarix ramosissima* Ledeb. came from deep soil water and groundwater because 70% of its absorbent roots were distributed in the 200–300 cm soil layer. The results from Liu et al. [36] suggest that the mean depths of water taken up by a *Populus euphratica* Oliv. forest were different and that *P. euphratica* trees access relatively deeper soil moisture as they age. However, the water use efficiency of the *P. euphratica* trees tended to decrease with age.

## 5. Conclusions

The mean soil depth of the water taken up by the walnut trees gradually decreased as the phenological stages of walnut advanced. The proportions of various soil layer water contributions to

plants differed throughout the growing season, and the proportions of deeper soil water contributions gradually increased as the phenological stages of walnut advanced. In addition to soil moisture, soil temperature may also be an important factor affecting root absorption depth. In the management of fruit trees, we can control the water absorption depth of roots by changing soil temperature. This study reports a useful method for identifying plant water sources, and the findings are of great significance for future irrigation management.

**Author Contributions:** Y.L. and G.Q. conceived and designed the experiments. Y.L. and S.Z. performed the experiments. Y.L., S.Z., and X.Z. analyzed the data. Y.L. and G.Q. wrote the manuscript; other authors provided editorial advice.

**Funding:** This research was funding by the nonprofit industry research subject of the State Forestry Administration in China (Grant No. 201504408) and the Science and Technology Project of Hebei Province in China (Grant No. 16236810D).

**Acknowledgments:** We appreciate the anonymous reviewers and the editor for thoughtful feedback that improved the manuscript.

**Conflicts of Interest:** The authors declare no conflict of interest.

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
