# Peer review of "The Depth of Water Taken up by Walnut Trees during Different Phenological Stages in an Irrigated Arid Hilly Area in the Taihang Mountains"

_forests, doi:10.3390/f10020121_

Round 1
Reviewer 1 Report
The current version of the manuscript has improved in readability and the presentation of results is now clearer, including both 18O and 2H and providing the necessary details about the estimation of water uptake proportions. However, I am concerned with some issues in the data presented:
1) Comparing the previous and current version of the manuscript, d2H profiles in figure 5 do not look exactly the same as the ones presented in the first version. Please clarify.
2) According to the soil type, how far are the values in figure 3 from saturation. Near 50% water content is quite high. Besides, some more detailed description of the soil would be useful. Ideally water holding capacity of % WC at field capacity and wilting point would help to understand the context.
3) I would also consider irrigation water as a potential source in MixSIAR, since it is not clearly reflected in the soil water. This would account by the water that is rapidly taken by the plant during irrigation, and that it not stored in the soil. Indeed, stem values in the biplot (figure 4) appear to be slightly drifted towards the lower d2H of the irrigation water, as compared to soil values.
Other comments:
Check carefully spelling and punctuation. There are several missing 'space' between words, and also changes in font size.
Figure 4 To avoid confusing the reader, the biplot in figure 4 should not use inverted axis. At first glance I thought there was some weird fractionation with soil samples, until I realized that the lower values were in the up-right corner.
line 281: 'Our research found that there was no clear difference in shallow soil water between the sprouting and leaf expansion stage and the blossoming and fruit-bearing stage, but the proportion of shallow soil water contributions to the walnut trees varied greatly.' I guess the authors refer here to 'soil water content' (please specificy). Although (overall) differences may not be statistically significant, in figure 3 there is a decreasing trend in shallow WC during the growing season. Also Figure 5 indicates that the evaporation increases in depth during the growing season. In this regard, I would soften the conclusions regarding the role of temperature. The results suggest that it might be relevant, but are not fully conclusive.
lines 299-306. This part should be omitted from the conclusions, it is mainly a summary of results.
Author Response
Thank you for your comments. Here are my answers to some questions.
Point 1: 1)Comparing the previous and current version of the manuscript, d2H profiles in figure 5 do not look exactly the same as the ones presented in the first version. Please clarify.
Response 1: I can't get the result with simultaneous input of hydrogen and oxygen data when I use IsoSources model to analyze the water contribution rate. That's why only hydrogen data were shown in the first manuscript. In the first review opinion, a reviewer advised me that with a multi-isotope approach and the application of most recent bayesian models (e.g. SIAR or MixSIAR) the study would provide more robust conclusions. So, the results using MixSIAR model were shown in this manuscript. When I rearranged my data, some of the data were rejected due to exceptions.
Point 2: 2)According to the soil type, how far are the values in figure 3 from saturation. Near 50% water content is quite high. Besides, some more detailed description of the soil would be useful. Ideally water holding capacity of % WC at field capacity and wilting point would help to understand the context.
Response 2: The Previous values of Fig.3 were directly measured by instruments. In response to your query, I contacted the instrument manufacturer. According to the manufacturer's instructions, I corrected the experimental data. New values of SWC are showed in the manuscript.
Point 3: 3) I would also consider irrigation water as a potential source in MixSIAR, since it is not clearly reflected in the soil water. This would account by the water that is rapidly taken by the plant during irrigation, and that it not stored in the soil. Indeed, stem values in the biplot (figure 4) appear to be slightly drifted towards the lower d2H of the irrigation water, as compared to soil values.
Response 3: I couldn’t understand the suggestion.
Point 4: Check carefully spelling and punctuation. There are several missing 'space' between words, and also changes in font size.
Response 4: It’s my carelessness. Thank you for point out my mistake.
Point 5: Figure 4 To avoid confusing the reader, the biplot in figure 4 should not use inverted axis. At first glance I thought there was some weird fractionation with soil samples, until I realized that the lower values were in the up-right corner.
Response 5: Thank you for your suggestion, and figure 4 has been revised.
Point 6: line 281: 'Our research found that there was no clear difference in shallow soil water between the sprouting and leaf expansion stage and the blossoming and fruit-bearing stage, but the proportion of shallow soil water contributions to the walnut trees varied greatly.' I guess the authors refer here to 'soil water content' (please specificy). Although (overall) differences may not be statistically significant, in figure 3 there is a decreasing trend in shallow WC during the growing season. Also Figure 5 indicates that the evaporation increases in depth during the growing season. In this regard, I would soften the conclusions regarding the role of temperature. The results suggest that it might be relevant, but are not fully conclusive.
Response 6: L281-297 The part of this discussion has been modified. See the revised version for details.
Point 7: lines 299-306. This part should be omitted from the conclusions, it is mainly a summary of results.
Response 7: Thank you for your suggestion.
Reviewer 2 Report
L. 8-9 However, this manuscript does not provide the data of root distribution!
L. 22-23 This statement is not supported by presented results.
L.43 Dawson et al. ?
L. 75-77 How was weather information obtained? It seems inconsistent with Fig. 1.
L. 161 Is "Great difference" a significant difference? Are these data daily mean soil temperature? I am not sure the authors' intention how to use this data set. If we consider the effect of evaporation on the isotopic fractionation, daily maximum soil temperature is more important.
L. 176 The values of SWC are too high! There may be a problem with calibration of the soil sensor system.
Fig. 4 Why are the dD and dO relationships different between soil and rainfall?
Fig 6 Does 0.6% of contribution have any meaning?
L. 234 Discussion part is redundant and not well organized. This study was done under fairly artificial conditions. Cited references must be carefully chosen.
L. 236-253, 269-270 The dO of rainfall ranged from -2 to -10 (Fig. 4). The effect of different dO (and dD) of precipitation must be discussed.
L 256-257 Too many repetition!
L262 Scientific names describe in italics.
L278-280 Scientific names describe in italics.
L281-297 The part of this discussion is too speculative. Neither appropriate experimental setting nor effective statistical analysis was performed.
L310-312 This statement is not supported by presented results.
Author Response
Thank you for your comments. Here are my answers to some questions.
Point 1:L. 8-9 However, this manuscript does not provide the data of root distribution!
Response 1: My expression is indeed somewhat inappropriate and has been revised.
Point 2: This statement is not supported by presented results.
Response 2: This statement is my inference of the experimental data. There are no difference in water content among different soil layers, but the water absorption depth has changed dramatically. The difference of water uptake depth of walnut roots in different phenological periods is ultimately determined by the water uptake of roots in different soil layers in different seasons. The amount of water absorbed by roots depends on the number and water uptake ability of roots. The water uptake ability of roots was affected by soil environment of which soil Soil moisture and soil temperature are very important soil factors.
Point 3:L.43 Dawson et al. ?
Response 3: Thank you very much for pointing out my mistake, and it’s my carelessness.
Point 4:75-77 How was weather information obtained? It seems inconsistent with Fig. 1
Response 4: Weather information are multi-year averages, while Figure 1 is the 2017 meteorological data. The two do not conflict.
Point 5: L.161 Is "Great difference" a significant difference? Are these data daily mean soil temperature? I am not sure the authors' intention how to use this data set. If we consider the effect of evaporation on the isotopic fractionation, daily maximum soil temperature is more important.
Response 5: Here, I want to show that the soil temperature of different soil layers is different. The difference of water uptake depth of walnut roots in different phenological periods is ultimately determined by the water uptake of roots in various soil layers in different seasons. The amount of water absorbed by roots depends on the number and water uptake ability of roots. The water uptake ability of roots was affected by soil environment of which soil Soil moisture and soil temperature are very important soil factors.
Point 6: L.176 The values of SWC are too high! There may be a problem with calibration of the soil sensor system.
Response 6: The Previous values of Fig.3 were directly measured by instruments. In response to your query, I contacted the instrument manufacturer. According to the manufacturer's instructions, I corrected the experimental data. New values of SWC are showed in the manuscript.
Point 7: Fig. 4 Why are the dD and dO relationships different between soil and rainfall?
Response 7: Soil water comes from not only rainfall, but also irrigation water. First, the dD and dO relationships was different between irrigation water and rainfall. Soil moisture was also affected by evaporation. So the dD and dO relationships different between soil and rainfall.
Point 8:Fig 6 Does 0.6% of contribution have any meaning?
Response 8: I’m sorry that I made a mistake about the data and units because of my carelessness.
Point 9: L.234 Discussion part is redundant and not well organized. This study was done under fairly artificial conditions. Cited references must be carefully chosen.
Response 9: The sentences of L.236-255 has been adjusted.
Point 10:L 256-257 Too many repetition!;L262 Scientific names describe in italics.;L278-280 Scientific names describe in italics.
Response 10: It’s my carelessness. Thank you very much for pointing out my mistake.
Point 11: L281-297 The part of this discussion is too speculative. Neither appropriate experimental setting nor effective statistical analysis was performed.
Response 11: L281-297 The part of this discussion has been modified. See the revised version for details.
Round 2
Reviewer 1 Report
Overall, I acknowledge the effort made by the authors to improve the manuscript and data presentation. I am fully satisfied with the last changes performed in the manuscript, which made the results clearer and more consistent.
Regarding my suggestion about the use of irrigation as a potential source, I am sorry if my explanation was not clear. I would try to clarify it:
My point was that, among the sources used as input for the MixSIAR model, the authors could try an alternative model, in which, together with the different soil layers, the mean values of irrigation water are included as one additional source. The reasoning behind is to assess whether the irrigation water is rapidly used by the plants, even before imprinting an isotopic signal in the soil (see Tang and Feng 2001, Earth and Planetary Science Letters; in the same line, see a recent review on the so-called "two water worlds hypothesis": Berry et al. 2017, Ecohydrology). In case it brings new clues about seasonal changes in plant water uptake, the outcome of this model could be included as a supplementary information.
Author Response
Thank you very much for you advice.
I consulted the two documents you mentioned and understand your point. I also believe the situation you mentioned existed. Before imprinting an isotopic signal in the soil, the irrigation water maybe rapidly used by the plants.
However, I’m not sure if it's appropriate that the mean values of irrigation water are included as one additional source. First, Soil water content was relatively low before irrigation. After irrigation, much of the soil water comes from irrigation water. In the early stage of irrigation, although irrigation water was not completely converted into soil water, it is also in the soil. If the mean values of irrigation water are included as one additional source, It may lead to repeated calculation of irrigation water. This may result in huge errors in the proportion of water resources contributed. in addition, the situation you mentioned also existed when it rains. If the mean values of irrigation water are included as one additional source, the values of rainfall should be also included as one additional source.
Your proposal is very good, but it requires additional experiments. Before and after irrigation water imprint an isotopic signal in the soil, collect the soil samples and plant stem samples and compare with them. Later, I will conduct further experiments to verify this point of view. However, the key point of this experiment is to understand the main depth of water taken up by walnut trees during different phenological stages, in order to provide guidance for walnut production.